# Immortalization of Salivary Gland Epithelial Cells of Xerostomic Patients: Establishment and Characterization of Novel Cell Lines

**DOI:** 10.3390/jcm9123820

**Published:** 2020-11-25

**Authors:** Braxton D. Noll, Alexandre Grdzelishvili, Michael T. Brennan, Farah Bahrani Mougeot, Jean-Luc C. Mougeot

**Affiliations:** Carolinas Medical Center-Atrium Health, Charlotte, NC 28203, USA; braxton.noll@atriumhealth.org (B.D.N.); alexandre.grdzelishvili@atriumhealth.org (A.G.); mike.brennan@atriumhealth.org (M.T.B.); farah.mougeot@atriumhealth.org (F.B.M.)

**Keywords:** Sjögren’s syndrome, xerostomia, salivary gland, SGEC, immortalization, acinar, ductal, spheroid

## Abstract

Primary Sjögren’s Syndrome (pSS) is an autoimmune disease mainly affecting salivary and lacrimal glands. Previous pSS studies have relied on primary cell culture models or cancer cell lines with limited relevance to the disease. Our objective was to generate and characterize immortalized salivary gland epithelial cells (iSGECs) derived from labial salivary gland (LSG) biopsies of pSS patients (focus score > 1) and non-Sjögren’s Syndrome (nSS) xerostomic (i.e., sicca) female patients. To characterize iSGECs (*n* = 3), mRNA expression of specific epithelial and acinar cell markers was quantified by qRT-PCR. Protein expression of characterization markers was determined by immunocytochemistry and Western blot. Secretion of α-amylase by iSGECs was confirmed through colorimetric activity assay. Spheroid formation and associated alterations in expression markers were determined using matrigel-coated cell culture plates. Consistent mRNA and protein expressions of both epithelial and pro-acinar cell markers were observed in all three iSGEC lines. When cultured on matrigel medium, iSGECs formed spheroids, secreted α-amylase after β-adrenergic stimulation, and expressed multiple acinar cell markers at late passages. One iSGEC line retained adequate cell morphology without a loss of SV40Lt expression and proliferation potential after over 100 passages. In conclusion, our established iSGEC lines represent a viable model for salivary research due to their passaging capacity and maintenance of pro-acinar cell characteristics.

## 1. Introduction

Development of restorative therapies for salivary gland dysfunction has been significantly hampered by a lack of accessible and pertinent cell culture models. Salivary glands are made of several cell types, including the saliva-producing acinar cells, ductal cells modifying the saliva as it travels through the lumen, and myoepithelial cells mediating acinus contraction. Acinar cells commonly fail to regenerate following high-dose radiation therapy, aging, or from autoimmune exocrinopathy such as primary Sjögren’s syndrome (pSS) [1,2,3]. pSS is a chronic autoimmune disease characterized by a loss of salivary acinar cell function and a decline in acinar progenitor cell populations exacerbated by lymphocytic glandular infiltration.

Dysfunction of the salivary glands can lead to a reduction in salivary flow, thereby altering saliva composition, pH, and buffering capacity, causing the sensation of dry mouth (xerostomia) [2]. Cell culture models for pSS and xerostomia have been limited, leading researchers to rely on cancer cell lines or short-term cultures of primary salivary gland epithelial cells (SGECs) [4,5]. Moreover, the cell lines extensively used in salivary research (e.g., HSG, A253, NS-SV-AC) are of male origin [4,6,7,8]. These cell lines may not recapitulate the pathological processes occurring in pSS, a disease with a higher incidence in female patients [9].

Isolated SGECs have been shown to share ductal and pro-acinar characteristics and exhibit a moderate cell division capacity when grown in serum-free low-calcium media [5,10,11,12,13]. As with any primary cell culture model, their limited growth potential can impede reproducibility and applications including, but not limited to, high-throughput drug screening assays. In addition, pSS patient derived SGECs show a markedly reduced proliferation capacity in vitro [3]. The reduced growth capacity of pSS SGECs is mirrored in affected patients’ salivary glands where narrowed pools of progenitor cells are found with markedly shortened telomeres [3]. Overall, suboptimal cell culture models combined with the reduced growth ability of pSS SGECs have proven to be major obstacles in salivary research.

Diagnosis of pSS may require a labial salivary gland (LSG) biopsy for histologic analysis of lymphocytic aggregates within the tissue [14]. LSG biopsy tissue is a common explant source for culturing SGECs, but due to limited availability based on clinical access, their potential as a universal model is restricted. Culture of SGECs is typically performed using a dual media approach, where the initial explant medium contains 2.5% fetal bovine serum and is replaced with a second serum-free media to sustain passaging and growth [5,11]. Minor differences in culture conditions, including FBS concentration and time spent in explant medium can have a profound effect on salisphere (i.e., 3D spheroid) formation and passage number, as previously shown for parotid gland progenitor cells isolated from mice [15].

Primary SGEC cultures are comprised of cells with pro-acinar and/or ductal cell-like features depending on the growth substrate, presence of serum in medium, and the specificity of characterization markers used [5,10,11,12,13]. The previous work of Fujita-Yoshigaki et al. demonstrated the dedifferentiation of primary parotid acinar cells occurs in culture, which could explain the variety of ductal and acinar cell features expressed [16,17]. In addition, Jang et al. revealed that primary SGECs (known as phmSG) of ductal origin exhibited trans-epithelial resistance, expressed several acinar and epithelial cell markers (i.e., AQP5, SLC12A2, AMY1A), and secreted α-amylase upon β-adrenergic stimulation [5]. Overall, SGECs represent a suitable cell culture model to investigate salivary gland dysfunction in vitro but lack the unlimited growth potential needed for widespread use.

To our knowledge no reliable immortalized human primary SGEC lines derived from female patient’s LSG biopsies have been developed or made readily available to researchers. To address this gap, the objective of our study was to generate an in vitro cell culture model for the investigation of the pathophysiology of salivary gland disorders. Using an SV40 Large-T (SV40Lt) lentiviral vector, we generated and characterized immortalized SGECs isolated from LSG biopsies of pSS and xerostomic (sicca) patients. The SV40Lt antigen subunit inhibits both cell cycle regulators, pRB and p53, and prevent telomere shortening-induced cell senescence in some cell types [18,19]. We characterized immortalized SGECs (iSGEC) lines from two non-Sjögren’s syndrome (nSS) female patients (referred to as iSGEC-nSS1 and iSGEC-nSS2) and one pSS (referred to as iSGEC-pSS1) female patient, based on the expression of pro-acinar and epithelial cell markers using various molecular methods. iSGECs were grown on matrigel-coated plates to determine three-dimensional (3D) spheroid forming ability, along with the extent to which spheroids recapitulated acinus characteristics, such as differentiated myoepithelial and acinar cells.

## 2. Materials and Methods

### 2.1. Culture of Salivary Gland Cell Lines (SGCLs) and HeLa Cells

Salivary gland cell lines, HSG and HSY, and HeLa cells were cultured in DMEM supplemented with 10% FBS (VWR, Radnor, PA, USA) (37 °C; 5% CO_2_). A253 cells were cultured according to the provider’s (ATCC, Manassas, VA, USA) recommendations [20]. HMC-3A cells were cultured according to the providers’ recommendations [21]. Salivary gland cell lines (SGCLs) (HSG, HSY, A253, and HMC-3A) and HeLa cells were grown in T-75 tissue culture-treated flasks (Corning Life Sciences, Tewksbury, MA, USA) until 80–90% confluency. For experiments, SGCLs and HeLa cells were re-plated using 0.25% trypsin +0.053 mM EDTA (Corning Life Sciences, Tewksbury, MA, USA).

### 2.2. LSG Biopsies and Culture of Salivary Gland Epithelial Cells (SGECs)

Atrium Health institutional review board (Charlotte, NC, USA) approval (IRB Protocol #: 08-16-24E) was granted for this study and all patients gave informed consent. 

Anti-Ro (SSA) serum-negative (-) xerostomic patients undergoing a LSG biopsy for the assessment of pSS according to the 2016 ACR/EULAR classification criteria were asked to participate in this study [6]. Xerostomic nSS and pSS patient clinical characteristics are listed in Table 1. Remaining LSG biopsy tissue was transported to the research laboratory in complete SGEC explant medium supplemented with 5x antibiotic/antimycotic solution (Gibco, Thermo Fisher Scientific, Waltham, MA, USA).

SGEC cultures were established according to the methods outlined by Jang et al. and are briefly explained as follows [3]. LSG tissue was minced into 0.5–1 mm^2^ fragments and placed in T-75 flask with 3.5 mL of complete SGEC explant medium supplemented with 1x antibiotic/antimycotic solution (Gibco, Thermo Fisher Scientific, Waltham, MA, USA) and grown (37 °C, 5% CO_2_). Complete SGEC explant medium consisted of 1:3 Ham’s F12 and DMEM supplemented with 2.5% FBS (VWR, Radnor, PA, USA), 20 μg/mL EGF (Gibco, Thermo Fisher Scientific, Waltham, MA, USA), 200 μg/mL insulin (MP Biomedicals, Santa Ana, CA, USA), 100 ng/mL hydrocortisone (Sigma-Aldrich, Saint Louis, MO, USA), and 1x antibiotic/antimycotic solution (Gibco, Thermo Fisher Scientific, Waltham, MA, USA). After 72 h, 5 mL of SGEC explant media was added. Cells at 80–90% confluency were split 1:3 into T-75 flasks with 0.05% trypsin +0.045 mM EDTA. Trypsin was neutralized by soybean trypsin inhibitor (Gibco, Thermo Fisher Scientific, Waltham, MA, USA) at a 1:1 ratio (*v*/*v*). For remaining passages, SGEC sub-culturing medium was used. SGEC sub-culturing medium consisted of Epi-life Basal^TM^ medium (0.06 mM Ca^2+^) (Gibco, Thermo Fisher Scientific, Waltham, MA, USA, Catalogue# MEPI500CA) supplemented with 1x human keratinocyte growth supplement (HKGS) (Gibco, Thermo Fisher Scientific, Waltham, MA, USA, Catalogue# S0015). Fibroblasts were gradually removed from culture using a combination of 0.02% EDTA or 0.01% trypsin.

### 2.3. Transduction of SGECs by Lentiviral SV40Lt Particles

During passage number 2 (p-2), SGECs were spit into 6-well (VWR, Radnor, PA, USA) tissue culture-treated plates at a confluency of 60–70% and allowed to adhere for 24 h. Cell medium was removed and replaced with 1 mL SV40Lt lentiviral supernatant (ABMgood, Richmond, Canada, Catalogue# G258) diluted with 1 mL of SGEC sub-culturing medium (2 mL total) and polybrene (Sigma-Aldrich, Saint Louis, MO, USA) at a final concentration of 4 μg/mL. After 48 h, SV40Lt lentiviral cell medium was replaced with SGEC sub-culturing medium and grown for 24 h. Medium was removed and cells were supplied with 1 mL SV40 lentiviral supernatant diluted with 1 mL of SGEC sub-culturing media (2 mL total) and polybrene at a final concentration of 4 μg/mL, for 48 h.

SV40Lt lentiviral cell medium was removed and replaced with SGEC sub-culturing medium and cells were grown to 80–90% confluency before being split into 6-well tissue culture-treated plates at a ratio of 1:6. For cell passages after SV40Lt lentiviral transfection, cells were split 1:6 up to passage 10. The remaining passages were split at 1:3 using T-75 (Corning Life Sciences, Tewksbury, MA, USA) tissue culture-treated flasks. To determine increased expression of pro-acinar markers in high Ca^2+^ supplemented medium, all iSGECs at early passage 14 (p-14) and only iSGEC-nSS2 at passage 80 (p-80) were subjected to 1.2 mM Ca^2+^ in SGEC medium for 72 h prior to experimentation [3].

### 2.4. RNA Extraction and cDNA Synthesis

RNA was extracted from iSGECs, SGCLs, and Hela cells in 6-well plates using the RNAeasy kit (Qiagen, Valencia, CA, USA) according to the manufacturer’s protocol. RNA was quantified using a Nanodrop 1000 (Thermo Fisher Scientific, Waltham, MA, USA). A total of 500 ng RNA from each sample was used per cDNA synthesis reaction. cDNA synthesis was carried out using the SmartScribe reverse transcription kit (Takara Bio, Mountain View, CA, USA) with random hexamers (20 μg) (Promega, Madison, WI, USA) and dNTPs (10 mM final) (Promega, Madison, WI, USA) according to the manufacturer’s protocol. After synthesis, the cDNA was diluted to a final volume of 200 µL.

### 2.5. Matrigel Induced 3D Spheroid Cultures

Matrigel (Corning Life Sciences, Tewksbury, MA, USA) diluted in 1.2 mM Ca^2+^ SGEC sub-culturing medium (2 mg/mL final concentration) was solidified in 8-well Nunc™ Lab-Tek™ II Chamber Slides™ (Thermo Fisher Scientific, Waltham, MA, USA) or 12-well tissue culture-treated plates (VWR, Radnor, PA, USA) and incubated for 1 h at 37 °C. Cells were seeded at 3 × 10^5^ or 1 × 10^6^ per well, respectively. Cell cultures on matrigel-coated plates were grown at 37 °C and 5% CO_2_ for up to 7 days.

### 2.6. Real-Time Quantitative and Semi-Quantitative Polymerase Chain Reaction

All qRT-PCR and semi-qRT-PCR reactions were performed using a BioRad C1000 Touch Thermal Cycler (Biorad, Hercules, CA, USA) in 96-well qPCR plates. All qRT-PCR reactions were carried out in triplicate wells per plate using three separate plates and average cycle threshold (Ct) values were used for quantification. Relative expression was calculated applying the ΔCt method for iSGECs, SGCLs, and Hela cells normalized to the expression of GAPDH. Per well, 10 μL of RT2 SYBR Green Fast Master Mix (Qiagen, Valencia, CA, USA) was added to 2 μL of cDNA, 1 μL of forward and reverse primers (10 mM) (IDT, Newark, NJ, USA) and 7 μL of nuclease-free water. Primers are listed in Appendix A along with their respective target for cell expression characterization purposes.

Semi-qRT-PCR was employed to detect the expression of the SV40Lt transcript at multiple passage numbers (Appendix A). Products from semi-qRT-PCR reactions were run on a 1% agarose TAE gel alongside a 1 kb DNA ladder (New England Biolabs, Ipswich, MA, USA) for 35 min and visualized using a gel doc system (GE Healthcare Life Sciences, Chicago, IL, USA).

### 2.7. Expression of ZO-1, AQP5, and α-Amylase by Western Blot

iSGECs and SGCLs were plated at 80% confluency in 6-well tissue culture-treated plates and allowed to adhere for 48 h prior to experimentation. To generate whole-cell lysates, medium was removed from cells and the wells washed 3 times with DPBS. Protein was harvested from cells using M-PER buffer (Pierce, Thermo Fisher Scientific, Waltham, MA, USA) and briefly sonicated before centrifugation at 15,000× *g* for 10 min. The resulting insoluble pellet was discarded, and the supernatant was used for Western blotting of ZO-1, AQP5, and vinculin.

For the determination of α-amylase secretion, iSGECs were grown in SGEC sub-culturing media without HKGS for 24 h and then replaced with complete SGEC sub-culturing media supplemented with 10μM epinephrine (MP Biomedicals, Santa Ana, CA, USA). After 45 min, the cell culture media were harvested and centrifuged (4 °C, 1000× *g*) for 10 min to remove whole cells. The media were then spun in Millipore concentrator tubes with a 10 kDa size exclusion limit (Millipore, Burlington, MA, USA) for 30 min and an appropriate amount of 6x reducing Lameli buffer (Roche, Basel, Switzerland) added to the resulting concentrated supernatant. iSGECs adhered to the tissue culture plate were lysed and harvested using M-PER buffer. Protein samples in M-PER were briefly sonicated and centrifuged at 15,000× *g* for 10 min and the pellet discarded.

Samples were measured by Bradford assay (Pierce, Thermo Fisher Scientific, Waltham, MA, USA) and equal amounts of protein were loaded into each lane and subjected to electrophoresis on a 7–14% gradient pre-cast SDS–PAGE gel (Biorad, Hercules, CA, USA). Proteins were transferred onto nitrocellulose membranes (GE Healthcare Life Sciences, Chicago, IL, USA) and blotted (12 h, 4 °C) with selected antibodies at listed concentrations (Appendix A) followed by incubation (room temperature, 1 h) with HRP-conjugated anti-mouse secondary (Cell signaling Technology, Danvers, MA, USA).

Membranes were washed (3 times, 5 min) and developed using Super Signal West pico Chemiluminescent substrate kit (Pierce, Thermo Fisher Scientific, Waltham, MA, USA). Western blots were photographed using an ImageQuant LAS4000 (GE Healthcare Life Sciences, Chicago, IL, USA) system. Densitometry was performed using Image Studio V.5.2 software (LI-COR, Lincoln, NE, USA) and protein levels were normalized to vinculin in their corresponding whole-cell lysate.

### 2.8. β-. Adrenergic Stimulation and Measurement of α-Amylase Activity in Supernatant

iSGECs were plated at a density of 4 × 10^5^ cells in either uncoated (2D) or coated (matrigel) tissue culture plates. Cells were grown in SGEC sub-culturing medium supplemented with 1.2 mM Ca^2+^ for 72 h or 5 days prior to experimentation. At the times indicated, medium supplemented with 10 μM epinephrine (MP Biomedicals, Santa Ana, CA, USA) was added. After 45 min, cell culture medium was collected and subjected to the colorimetric amylase activity assay (Biovision, Milpitas, CA, USA) as per the manufacturer’s protocol.

### 2.9. Immunocytochemistry (ICC)

iSGECs and SGCLs were plated onto Coating Matrix (2D) (Gibco, Thermo Fisher Scientific, Waltham, MA, USA,) covered 8-well Nunc™ Lab-Tek™ II Chamber Slides™ (Thermo Fisher Scientific, Waltham, MA, USA). Antibodies and concentrations are listed in Appendix A. Cells were grown for 48–72 h, then fixed with ice-cold methanol (−20 °C, 10 min) and incubated with the following antibodies directed against them: KRT8, KRT18, KRT19, AQP5, ZO-1, α-SMA, Vimentin, and E-cadherin proteins. Cells were also fixed with 4% paraformaldehyde (room temperature, 15 min) to stain Ki-67 and α-amylase. Fixed cells were permeabilized with 0.25% Triton X-100 in TBS (room temperature, 10 min).

Cells were fixed then washed (3x) in TBS and incubated (room temperature, 5 min) in 3% H_2_0_2_ (*v*/*v*) TBS to block endogenous peroxidases. Next, cells were incubated in blocking buffer consisting of 1% BSA (*w*/*v*), 2% skim-milk (*w*/*v*), and 0.1% Tween-20 in TBS (room temperature, 1 h). Primary antibodies were diluted in TBS with 0.1% Tween-20 and 1% BSA. The fixed cells were incubated with the primary antibody (4 °C, 12 h) (Appendix A). Cells were washed with TBS (3 times, 5 min each) and incubated with anti-mouse secondary antibodies at different concentrations (Appendix A) and diluted in TBS with 0.1% Tween-20 (room temperature, 1–2 h). After incubation, cells were washed with TBS (3X).

Monolayer culture-fixed cells were incubated in diaminobenzidine (DAB) (Pierce, Thermo Fisher Scientific, Waltham, MA, USA) 1x solution (2–7 min). DAB was washed away using TBS to stop the reaction and the cells were counterstained with Gills II hematoxylin (Richard-Allan Scientific, Kalamazoo, MI, USA) (diluted 1:5 *v*/*v* DPBS) for 30 s to highlight nuclei. Slides without primary antibody served as negative controls.

For immunocytochemistry (ICC)–immunofluorescence (IF) visualization of 3D spheroid cultures grown on matrigel, cells were fixed using 4% paraformaldehyde (room temperature, 25 min) and permeabilized with 0.25% triton X-100 in TBS (room temperature, 10 min). The same blocking and incubation procedure previously outlined was followed except 4′,6-diamidino-2-phenylindole (DAPI) (Abcam, Cambridge, UK) was used to highlight DNA. Slides without primary antibody served as negative controls.

Slides were observed using an Olympus BX51 fluorescence microscope (Shinjuku City, Tokyo, Japan) and photographed with an Olympus DP70 (Shinjuku City, Tokyo, Japan) mounted camera. Cells in culture plates were viewed by phase contrast on an Olympus IX-71 inverted fluorescence microscope and photographed using an Olympus DP70 mounted camera. Proliferation scores were determined based on Ki-67 protein detection, a nuclear protein expressed during several cell cycle phases (G1/S/G2/M), which is frequently employed as an indicator of actively proliferating cells [22]. Proliferation scores (%) were generated by selection of three random images for nuclei counting using Fiji-ImageJ software [23]. The total number of cell nuclei was divided by the total number of Ki-67 (+) nuclei and averaged among three images.

## 3. Data Analysis and Statistics

Data are presented as the mean +/−standard error of the mean (SEM) from a minimum of three separate experiments. Significant results were calculated using Student’s *t*-test on GraphPad Prism 8 (San Diego, CA, USA). *P*-values were determined to be significant using the Holm–Šidák post hoc test for multiple comparisons with alpha = 0.05.

## 4. Results

### 4.1. Primary Isolation, Culture, and Growth of iSGECs

LSG biopsy fragments remained in SGEC explant medium for approximately 7–14 days to ensure sufficient cell outgrowth. Once the cells reached approximately 70–80% confluency, the culture medium was replaced with SGEC sub-culturing medium. Transitioning to the serum-free low calcium (0.06 mM) medium for 3–5 days prior to trypsinization reduced fibroblast contamination in later passages. Before SV40Lt transduction in all three cell lines, two separate populations of cells were observed and consisted of either larger cells with complex cytoplasm or smaller cells with reduced cytoplasm. The smaller cells were predominant during later passages in all three iSGEC lines. At late passages iSGEC-pSS1 gave rise to colonies appearing that were mixed in size, with polygonal or cobblestone-like cells with differing rates of cytoplasmic complexity (Figure 1A). iSGEC-nSS1 late passages demonstrated a mix of small polygonal and filiform-appearing cells (Figure 1B). Mixed morphology of iSGECs most likely indicates an inhomogeneous population (Table 2) (Figure 1A–D). Therefore, we selected iSGEC-nSS2 for late-passage outgrowth and in-depth characterization over iSGEC-pSS1 and -nSS1. Long term passaged iSGEC-nSS2 exhibited a stable, small cobblestone type of morphology and did not give rise to filiform-appearing cells (Figure 1C,D). iSGEC-nSS2 cultured in 1.2 mM Ca^2+^ acquired more complex appearing cytoplasm with granulations and formed tight clusters at both early and late passages (Figure 1E,F).

During later passages (iSGEC-nSS2, p-80), colony formations that were different in their overall shape arose from single-cell clonal outgrowths (Appendix A). However, these single-cell outgrowth populations quickly became incorporated into other colonies and were unremarkable at lower confluency.

Proliferation rates of iSGEC-nSS2 did not substantially waver across early (p-14) and late (p-80) passages, as indicated by Ki-67 staining percentage (Figure 1G). No decrease in the % of Ki-67 (+) cells was observed at later passages. Spheroid formation on matrigel was observed after 24 h in all three cell lines (Figure 2, Appendix A). iSGEC-nSS2 retained its spheroid formation ability into late passages (>p-80), which could be observed after 24 h of plating. At three days in matrigel, spheroids measured roughly ~75–200 µm in all iSGECs.

### 4.2. Characterization of iSGECs by qRT-PCR

mRNA expression of epithelial and acinar markers was determined by qRT-PCR in iSGECs, SGCLs, and Hela cells. Expression of epithelial and acinar cell markers showed slight deviation among all three iSGECs when grown as a monolayer culture (Figure 3A,B). Compared to SGCLs and Hela, A253 and HMC-3A showed the greatest similarity to iSGEC cultures. Mucoepidermoid carcinomas have been shown to exhibit increased expression of AQP5 and among all cell lines analyzed, HMC-3A expressed AQP5 highest (Figure 3A) [24].

The two consistently over-expressed pro-acinar genes by iSGEC-nSS2 were ANO1 and SLC12A2, compared to both iSGEC-pSS1 and -nSS1 (Figure 3A). We did not observe a difference in the pro-acinar expression of AMY1A, AQP5, STIM1, and STIM2 when comparing iSGEC-nSS2 to both iSGEC-pSS1 and -nSS1.

iSGEC-nSS1 exhibited the highest expression of both VIM and CST3, and the lowest expression of KRT19 (a ductal/epithelial cell maker of salivary glands) among iSGECs, indicating a heterologous mesenchymal phenotype (Figure 3B) [5,25]. KRT5 is a marker for progenitor and basal ductal cells of the salivary gland [26,27]. In agreement with the work by Jang et al. on primary SGECs, KRT5 was the gene with highest expression in iSGECs by qRT-PCR among the characterization markers, which did not differ significantly among all three iSGEC lines [5]. HSG, HSY, and HeLa cells expressed most transcripts consistently at the same level and exhibited a mesenchymal expression pattern reflected by iSGEC-nSS1 with low CDH1 and high VIM expression [25].

### 4.3. Changes in mRNA Expression among Early and Late Passage iSGECs by qRT-PCR

Expression of acinar and other characterization markers was assessed over early and late passages. iSGEC-pSS1 was the most stable cell line where only AMY1A and CST3 expression increased in later passages. Both iSGEC-nSS1 and -nSS2 exhibited pro-acinar changes in expression profiles at later passages where most markers increased (Figure 4A). In iSGEC-nSS1 p-45, pro-acinar markers (AMY1A, AQP5, ANO1, SLC12A2, CST3, and TRPC1) increased in expression compared to early (p-14) cultures (Figure 4A). iSGEC-nSS2 at p-80 expressed AMY1A, ORAI1, STIM1, SLC12A2, CST3, and TRPC1 higher, and only one marker lower (ANO1) compared to p-14. Both AQP5 and STIM2 were stably expressed during extended passaging in iSGEC-nSS2. Overall, AMY1A and CST3 increased during late passages of all three iSGECs lines, which could be important for maintaining pro-acinar characteristics.

Characterization markers were the most differentially expressed and exhibited the greatest changes in late passaged iSGECs. Epithelial marker CDH1 was the only stably expressed gene among all characterization markers, whereas ZO-1 increased in late passages of all three iSGECs (Figure 4C,D). iSGEC-nSS2 maintained an epithelial expression pattern, including a decrease in the mesenchymal marker VIM and a significant increase in CLDN1 expression. Over passages, both KRT5 and KRT19 decreased in expression and the progenitor cell marker NANOG increased and could indicate a shift towards a less-differentiated cell population in iSGEC-nSS2.

### 4.4. Effects of Ca^2+^ on iSGECs mRNA Expression

Calcium concentrations in media affect SGEC expression of pro-acinar genes, including AQP5, ORAI1, STIM1 and STIM2 [5,28]. Jang et al. demonstrated an increase in the activity of the store-operated Ca^2+^ entry (SOCE) system that upregulates AQP5 expression through the nuclear factor of activated T-cells 1 (NFAT1) [28]. To evaluate changes produced by an increase in Ca^2+^, we cultured iSGECs in 1.2 mM Ca^2+^ for three days and then assessed mRNA expression by qRT-PCR. Early iSGEC-nSS2 cultures (p-14) did not exhibit any alterations in expression of acinar markers when cultured in 1.2 mM Ca^2+^ (Figure 5A,B). Late iSGEC-nSS2 (p-80) cells supplemented with 1.2 mM Ca^2+^ demonstrated an increase in AQP5 and ANO1 mRNA expression and decreases in other acinar markers (AMY1A, ORAI1, STIM1, SLC12A2, TRPC1) (Figure 5A,B). Moreover, a change in ductal characterization marker KRT19 was decreased in p-14 iSGEC-nSS2 cells cultured in 1.2 mM Ca^2+^ (Figure 5C,D). Late (p-80) iSGEC-nSS2 responded differently via an increased expression of tight junction component ZO-1 and myoepithelial marker α-SMA. The lack of response to 1.2 mM Ca^2+^ in iSGC-nSS2 p14 cells reiterates the possibility of a de-differentiated cell population seen in Figure 4A,B, where p-80 demonstrated a response to increased Ca^2+^ that was more similar to other SGEC publications [5,28].

### 4.5. Characterization of iSGECs by ICC and Western Blot

#### 4.5.1. ICC and Western Blot of iSGECs in Monolayer Culture

The cytokeratins KRT8, KRT18, and KRT19 are cytoplasmic proteins expressed in the salivary gland epithelium and are characterization markers for ductal and acinar cells [29,30]. We first characterized iSGECs by KRT8/18/19 protein expression and found iSGEC-nSS1 expressed all three epithelial markers more intensely and uniformly than iSGEC-pSS1 and -nSS2 (Table 2, Appendix A). KRT19 protein localized within cells containing a larger cytoplasm and were dispersed throughout iSGEC-nSS2 cultures, whereas smaller cells demonstrated a low expression or a lack thereof (Appendix A). KRT18 was the most uniform and ubiquitously expressed cytokeratin among iSGECs, whereas KRT8 exhibited focal protein expression in iSGEC-nSS2 (Appendix A). Heterogenous expression of KRT19 and KRT8 could indicate multiple stages of differentiation among cells [26]. Overall, the expression of KRT8, KRT18, and KRT19 indicates these cell lines to be of ductal origin and reflects other long-term cultures of SGECs in both mice and humans [5,30,31].

Acinar cell markers, AQP5 and AMY1A, were expressed in all three iSGECs, and highest in both iSGEC-pSS1 and -nSS2. Within these cultures, cells in close contact expressed the highest levels of AQP5, unlike AMY1A where intercellular contact or proximity to other cells did not appear to have an influence. We observed an increase in AQP5 protein expression in both p-14 and p-80 iSGEC-nSS2 cultures when supplemented with 1.2 mM Ca^2+^. Increased AQP5 protein was determined by ICC and replicated by Western blot using whole-cell lysates (Appendix A, Figure 6A). ZO-1 expression was low in all iSGECs when accessed by ICC and confirmed by Western blot in whole-cell lysates. iSGEC-nSS1 exhibited the lowest expression of ZO-1 and CDH-1 and higher expression of VIM in monolayer culture, which reiterates the observed mesenchymal phenotype [25].

In the case of iSGEC-nSS2, α-SMA was not detected in either p-14 or p-80 monolayer cultures as compared to iSGEC-pSS1 and -nSS1, where it was expressed at very low levels (Appendix A). Pringle et al. demonstrated low α-SMA expression in cultured salivary gland stem cells (SGSCs) likely originating from intercalated and striated ductal cells, which could explain these results [3]. Vimentin is a marker for mesenchymal cells within the salivary glands [32,33]. Vimentin was expressed sporadically in iSGEC-nSS2 monolayer culture and could indicate a small population of mesenchymal cells or epithelial cells reverting to a mesenchymal-like state within cultures (Appendix A).

In all three iSGEC lines, α-amylase was secreted into the cell culture medium after β-adrenergic stimulation by epinephrine (Figure 6B). Additionally, iSGEC-nSS2 early (p-14) and late passages (p-80) secreted α-amylase into cell culture media with and without β-adrenergic stimulation (Figure 6B). iSGECs-nSS2 p-80 secrete α-amylase, demonstrating retention of pro-acinar cell characteristics after significant cellular expansion. Moreover, iSGEC-nSS2 cells cultured on matrigel formed spheroid structures and secreted α-amylase in response to 10 mM epinephrine stimulation (Figure 6C). An overall increase in α-amylase secretion when compared to unstimulated cells was observed in iSGEC-nSS2 and could indicate a stricter regulation of secretion within the differentiated cells.

#### 4.5.2. 3D-Matrigel Spheroid Culture

Matrigel contains a mixture of basement membrane and extracellular matrix proteins, which have been shown to promote SGEC and salivary gland stem cell (SGSC) differentiation [3,11,34,35]. SGECs plated on matrigel form spheroids resembling a basic acinus structure, composed of differentiated acinar and ductal cells [3,34,35]. iSGECs began to form spheroids within 24 h of plating on matrigel (2 mg/mL) and were cultured for a total seven days. ICC–IF protein expression analysis of iSGEC-nSS2 spheroid cultures showed high localization of acinar and epithelial cell markers ZO-1, AQP5, and AMY1A within acinus-like spheroid structures (Figure 7). Cytokeratins KRT8, KRT18, and KRT19 were all expressed in iSGEC-nSS2 cultured on matrigel (Figure 7). We observed greater levels of KRT8 expression within interior cells of spheroids similar to previously described high expression in luminal cells [36]. At a matrigel concentration of 2 mg/mL, some cells formed monolayer sheets across the matrigel that expressed KRT18 and KRT19 at high levels (Figure 7). α-SMA protein expression was localized to myoepithelial-like cells located on the spheroid exteriors.

mRNA expression analysis by qRT-PCR of iSGEC-nSS2 matrigel cultures demonstrated an increase in pro-acinar cell expression of AQP5 compared to monolayer cultures (Figure 8A,C). Spheroids from late passaged iSGEC-nSS2 expressed AMY1A and AQP5 at levels higher than early (p-14) cultures while expressing ANO1 at significantly lower (Figure 8D). Most characterization markers in iSGEC-nSS2 (p-14 and p-80) cultured on matrigel were differentially expressed (Figure 8E–G). Two consistent differentially expressed genes were VIM and KRT19, where VIM increased in matrigel cultures and KRT19 decreased. Similar to monolayer cultures on plastic, p-80 expressed NANOG at higher and KRT19 at lower levels, indicating a potential dedifferentiation when cultured for extended periods. However, p-80 exhibited the same increase in AQP5 mRNA expression as p-14 iSGEC-nSS2 cells on matrigel, retaining their ability to differentiate.

## 5. Discussion

This is the first study to establish immortalized primary cultures of xerostomic female patients using LSG biopsy tissue. We generated three novel composite iSGEC lines, with one capable of growing over 100 passages and maintaining expression of characterization markers.

Characterization of iSGEC-nSS2 indicates these cells to be of an undifferentiated ductal cell origin due to their high mRNA expression of KRT5 and relatively spotty distribution of KRT19 protein by ICC during monolayer culture on plastic [37,38]. When grown on matrigel, expression of KRT19 protein appeared more uniform over cells spread across the dish as a monolayer and is a marker expressed in both acinar and ductal cells [29,39]. Although a low concentration of 2 mg/mL was used for the entirety of this study, we tested whether higher at concentrations of matrigel, those monolayer cells would remain. At a concentration of (4.5 mg/mL) those monolayer cells fully incorporated into spheroid structures. 

Isolated acinar cells undergo dedifferentiation into ductal-like cells when explanted into cell culture plates [17]. Dedifferentiation of rat acinar cells can be halted by the inhibition of Src and p-38 pathways, allowing the cells to retain high cytoplasm complexity and size in culture [17]. Understanding the interactions between ductal and acinar cells within the salivary gland epithelium could provide more insight into acinar cell regeneration and population maintenance. Additionally, SGECs have demonstrated the ability to trans-differentiate into acinar-like cells in matrigel cultures [10,40]. Pathways regulating trans-differentiation would likely provide viable therapeutic targets, for example, with small-molecule inhibitors, bypassing the need for stem-cell implantation [41]. iSGEC-nSS2 spheroids resembled acinus-like structures with cells high in AQP5 expression and exterior surrounding myoepithelial-like cells expressing α-SMA (Figure 7). Changes in protein expression exemplified by a lack of α-SMA in iSGEC-nSS2 monolayer cultures and high expression in matrigel indicate cell differentiation during spheroid formation. 

The KRT5 transcript was found highly expressed in our iSGEC cultures similar to non-transformed SGECs, which is in agreement with reported protein expression by progenitor and ductal basal cells within the salivary gland epithelium [5]. Under normal homeostatic conditions, KRT5-positive cells give rise to intercalated and striated ductal populations [42]. However, during radiation-induced salivary gland injury KRT5-expressing ductal cells are capable of differentiation into acinar cells [42]. KRT5-positive ductal cells demonstrate higher robustness than acinar cells to radiation induced salivary gland damage and represent a favorable target in restoring acinar cell populations within damaged salivary glands [38]. Importantly, SV40Lt immortalization has been shown to not affect the differentiation potential of immortalized progenitor cells [43]. iSGEC-nSS2 could be a suitable model for understanding the cellular factors associated with trans-differentiation potential due to their high KRT5 mRNA expression.

Early (p-14) and late (p-80) passaged iSGEC-nSS2 cells expressed AQP5 at higher levels in spheroid cultures (Figure 8B,C), possibly indicating a greater extent of differentiation and therefore displaying increased heterogeneity. However, the expression of characterization marker KRT19 decreased while VIM expression increased in iSGEC-nSS2 cells cultured on matrigel. Higher VIM expression may indicate an increase in stemness properties by dedifferentiation, along with reduced KRT19 expression (differentiated ductal cell marker), which could also indicate an increase in stemness [26,44]. Similarly, changes in VIM and KRT19 expression were observed in long term 2D cultured (p-80) cells (Figure 4D). Overall, dedifferentiation of iSGEC-nSS2 cells in both 2D and 3D cultures is likely associated with their high growth and differentiation potentials when cultured on matrigel. Last, iSGEC-nSS2 could offer an in vitro alternative for salivary gland developmental research due to its dedifferentiation properties.

Direct introduction of an AQP1 expressing AAV vector into salivary glands demonstrated favorable clinical trial results where participants experienced a subjective decrease in xerostomia symptoms [45]. Although the original rationale behind AQP1 AAV therapy was to increase water permeability in ductal cells, it was later found that acinar cells were responsible for the expression of AQP1 and saliva secretion [46]. As previously stated, KRT5 protein expressing ductal progenitor cells are more abundant and can differentiate into acinar cells after injury [38]. Spheroid cultures obtained via iSGEC-nSS2 differentiation could serve as a suitable model to investigate how KRT5-positive progenitor cells possibly contribute to the generation of AQP1-expressing acinar after salivary gland damage and AQP1 AAV therapy.

## 6. Limitations

SGECs express AQP5 mRNA and protein under established culturing conditions, although being commonly identified of ductal origin [5,12]. Moreover, a spontaneously immortalized mouse cell line expressing mixed ductal and myoepithelial protein markers when cultured as a monolayer showed mRNA expression of acinar markers in monolayer and 3D culture [40]. Markers of differentiated ductal cells include KRT8 and KRT19 among others [26]. The lack of uniform expression by either KRT8 and KRT19 in monolayer cultures of iSGEC-nSS2 indicates a potential progenitor cell population, which is capable of differentiation into acinar-like and ductal-like cells when cultured on matrigel. Following culture on matrigel, iSGEC-nSS2 cells could be replated onto plastic tissue culture plates and further grown, thereby potentially expanding the differentiated cell populations for further analysis. To determine the extent of differentiation among acinar-like cells, immunocytochemistry targeting the terminally differentiated acinar cell marker, MIST1, should be employed. Moreover, dual staining of acinar and differentiated ductal cell markers to better identify and characterize the iSGEC-nSS2 cell line could be required to determine the extent of heterogeneity within the monolayer cultured cells.

Further research is needed to analyze the presence of tight junctions and their location throughout the apical portion of the plasma membrane, an important feature for polarized secretion by acinar cells [47]. Previous studies have demonstrated trans-epithelial resistance and polarized secretion of α-amylase within monolayer SGEC cultures [5]. Among the several cell mechanisms dictating epithelial polarization and directional secretion, tight junction formation is a critical factor [47]. Techniques better visualizing the location of tight junctions within iSGEC-nSS2 spheroids should be employed to better determine the extent of polarization and direction secretion of the high AQP5-expressing acinar cells. Furthermore, ZO-1 was expressed at low levels in monolayer iSGEC cultures indicated by immunocytochemistry and its presence confirmed by Western blot. Spheroids were uncut during the immunocytochemistry procedure, which could have impacted the interior binding of the ZO-1 antibody or the visualization of the 3D architecture.

When salivary glands are damaged, acinar cells revert to a ductal-like state in mouse models [48]. Similar to the phenomenon observed in mice, rat acinar cells when extracted and cultured on plastic revert to a ductal like state, which can be reversed by inhibition of p-38 and/or Src pathways [16,17]. Due to the outlined characteristics of iSGEC-nSS2, populations of differentiated acinar–ductal cells and ductal–progenitor cells could be present. Overall expression of AQP5 protein within iSGEC-nSS2 monolayer cultures localized within the plasma membrane and cytoplasm. Confirmation of AQP5 protein was further demonstrated by Western blot and exhibited a level of protein expression comparable to HMC-3A, which expressed AQP5 mRNA highest among all cell lines tested. When transitioned to culture on matrigel, AQP5 protein expression appeared to increase substantially within spheroid formations by ICC–IF. The increase in AQP5 expression among cells in matrigel culture could be a result of either de-differentiated acinar cell populations or progenitor-like cell populations (KRT5-positive) further differentiating into acinar cells. Before iSGEC-nSS2 use in developing therapies for increasing AQP5 expressing cells within the salivary gland, further cell sorting would need to be employed to ensure a homogenous cell population to the extent possible for reproducibility in drug assays.

In addition, age, gender, and associated hormonal changes account for possible factors affecting expression profiles of iSGECs, impacting their biological properties. Moreover, acinar cells of lacrimal glands, acinar progenitor cells of salivary glands, and saliva DNA of pSS patients may have shortened telomeres, which could be more prominent with older age [3,49,50]. Extensive cell passaging due to age combined with chronically shortened telomeres could lead to chromosomal instability and changes in DNA expression. Short Tandem Repeat (STR) profiling would be useful in the future to determine the extent of chromosome stability when passaged for extended periods and to establish a unique identifier for iSGEC-nSS2.

Although spheroids were formed by iSGEC-nSS2 in both early and late cultured cells, lumen formation needs to be further visualized, which could be useful in determining the extent of differentiation and cell types present. Furthermore, a panel of cell lines derived from different types of xerostomic patients (i.e., nSS “sicca” or radiation-induced) may be needed to develop effective treatments.

## 7. Conclusions

The iSGECs generated in our study represent a preliminary model system for the development of therapies targeting salivary gland dysfunction. We have demonstrated that iSGEC-nSS2 cells derived from a sicca female patient retained the ability to form spheroids with differentiated cell types at late passages (p-80) and exhibited a significant proliferation capacity (>100 passages) when cultured as a monolayer. Overall, iSGECs could be used as an alternative to currently available cell culture models in salivary gland research.

## Figures and Tables

**Figure 1 jcm-09-03820-f001:**
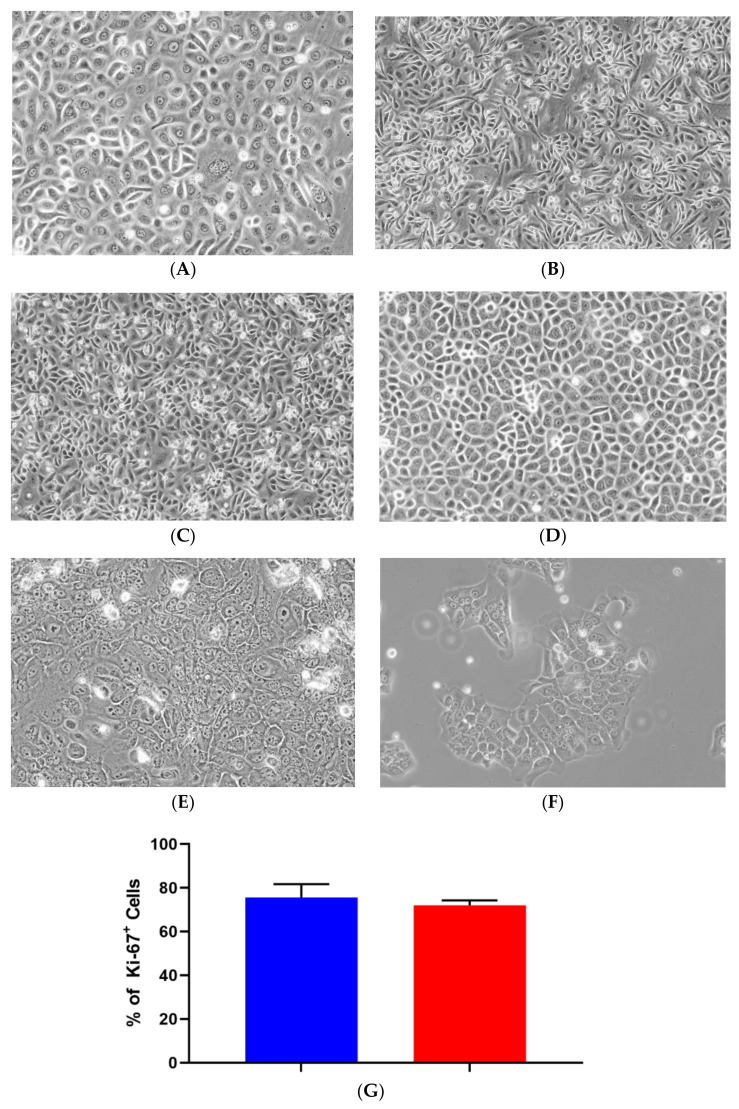
Morphology and proliferation of iSGECs. **Legend.** Representative images of early passage (p-14) monolayer culture (**A**) iSGEC-pSS1 (**B**) iSGEC-nSS1 (**C**) iSGEC-nSS2 and (**D**) late (p-80) iSGEC-nSS2. iSGEC-nSS2 “cobblestone-like” morphology was maintained across multiple passages when grown in SGEC sub-culturing media with low (0.06 mM) Ca^2+^. Cell morphology changes in (**E**) iSGEC-nSS2 p-14 and (**F**) iSGEC-nSS2 p-80 are observed when medium is supplemented with 1.2 mM Ca^2+^ (final concentration) for 72 h. (**G**) Proliferation of iSGEC-nSS2 was assessed by the percentage (%) of Ki-67^+^ cells using immunocytochemistry. No significant difference in the percentage of Ki-67^+^ cells was detected between early (p-14) and late (p-80) iSGEC-nSS2 cultures. Student’s *t*-test (* *p* < 0.05, NS = not significant). Data are means +/− SEM. Magnification ×20, scale bar = 200 µm.

**Figure 2 jcm-09-03820-f002:**
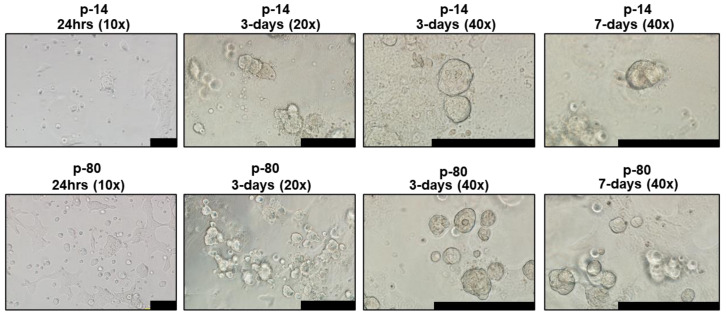
iSGEC-nSS2 early (p-14) and late (p-80) spheroid formation on matrigel. **Legend.** Cells were initially seeded at 4 × 10^4^ per well. Cultures began to form spheroids at 24 h on matrigel (2 mg/mL). Early passages (e.g., p-14) formed larger, but overall fewer spheroids when assessed after 3 days. Spheroids appeared to reach their final size after 3–4 days. Scale bars represent 200 µm.

**Figure 3 jcm-09-03820-f003:**
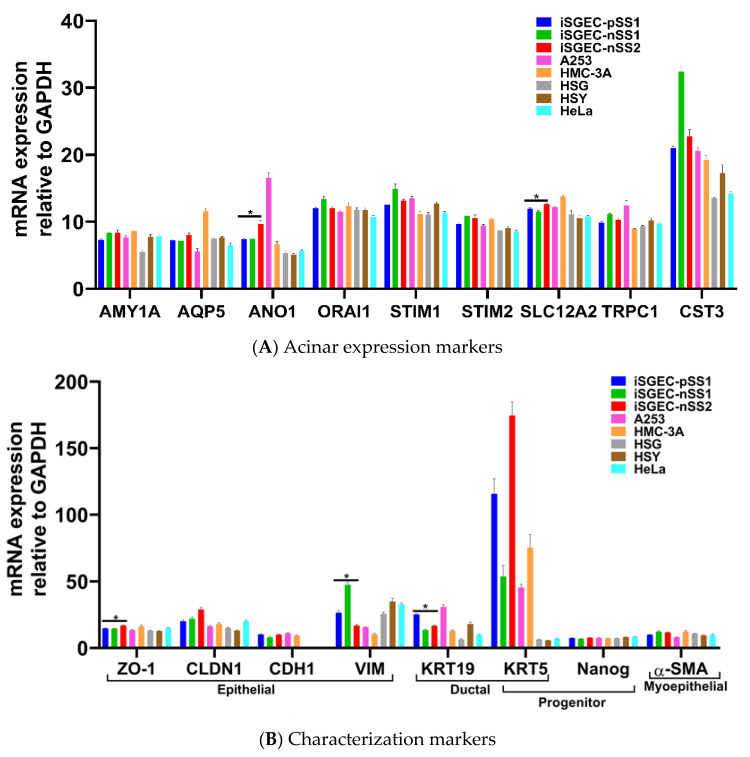
mRNA expression of acinar and characterization markers in monolayer cultured iSGECs, SGCLs, and HeLa cells by qRT-PCR. **Legend.** (**A**) Pro-acinar markers were highly expressed in all iSGECs. ANO1 and SLC12A2 were expressed highest in iSGEC-nSS2 compared to other iSGECs. (**B**) Characterization markers for epithelial, ductal, progenitor, and myoepithelial expression in all cell lines examined. The epithelial marker ZO-1 is an integral component of tight junctions and was overexpressed, whereas Vimentin (VIM) was under-expressed in iSGEC-nSS2. KRT-19, a ductal cell marker, was differentially expressed in iSGECs. * Indicates significant difference in expression determined by Student’s *t*-test (alpha = 0.05) and corrected using the Holm–Šidák method. Error bars represent mean +/− SEM.

**Figure 4 jcm-09-03820-f004:**
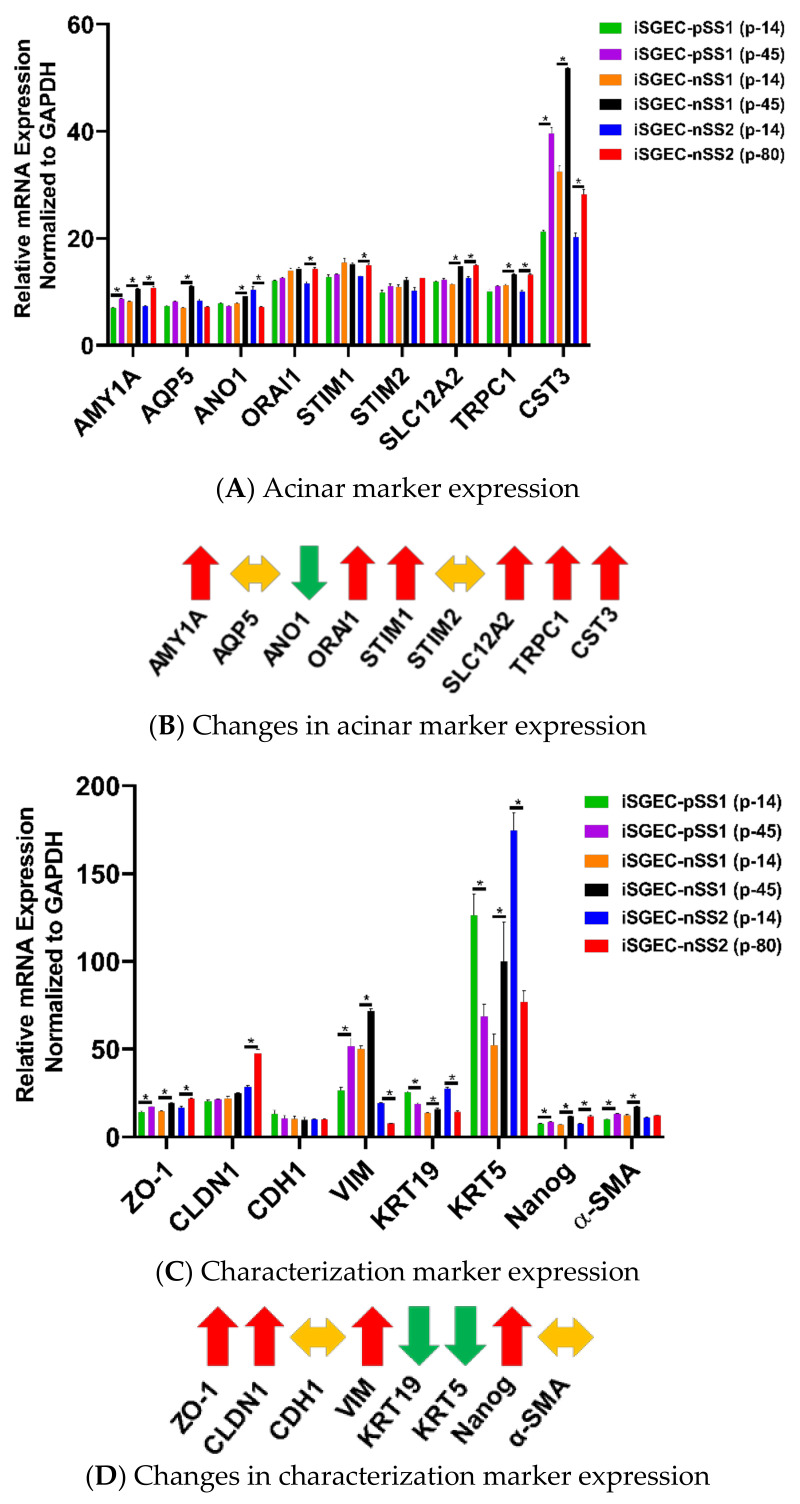
Changes in mRNA expression of early and late passaged iSGECs by qRT-PCR. **Legend.** mRNA expression of acinar (**A**) and characterization markers (**C**) in monolayer cultures of early and late passaged iSGECs by qRT-PCR (ΔCT). iSGECs increased in AMY1A (**A**), ZO-1 (**C**), and Nanog (**C**) in later passages. Changes in gene expression of iSGEC-nSS2 early (p-14) and late (p-80) are indicated with arrows: increased (**red**), decreased (**green**), and no change (**yellow**). (**B,D**) iSGEC-nSS2 differentially expressed several acinar and characterization markers in later passages. Significance of differences between the means was determined by Student’s *t*-test and *p*-values corrected using the Holm–Šidák multiple comparisons post hoc test (alpha = 0.05). Error bars represent mean +/− SEM. * *p* < 0.05.

**Figure 5 jcm-09-03820-f005:**
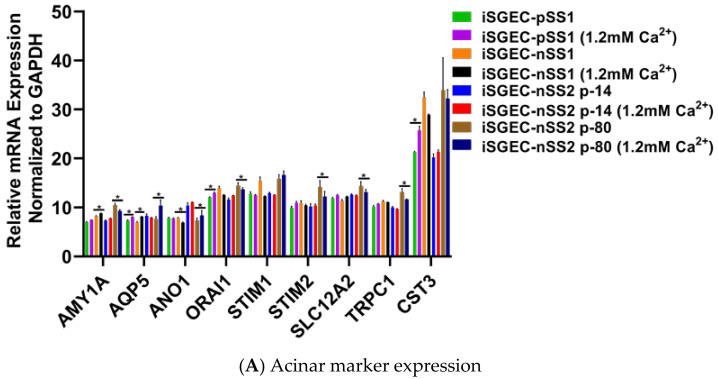
Changes in mRNA expression mediated by 1.2 mM Ca^2+^. **Legend.** Changes in acinar (**A**–**C**) and characterization markers (**D**–**F**) gene expression in iSGECs when cultured in 1.2 mM Ca^2+^ for 72 h. Changes in gene expression of iSGEC-nSS2 are indicated with arrows: increased (**red**), decreased (**green**), and no change (**yellow**). (**B**) iSGEC-nSS2 p-14 did not display changes in expression whereas late passaged (p-80) exhibited increases in acinar markers AQP5 and ANO1. AMY1A, ORAI1, STIM2, SLC12A2, and TRPC1 expression decreased in 1.2 mM Ca^2+^ p-80 cells. (**C**) Overall, characterization markers exhibited little change in 1.2 mM Ca^2+^ supplemented medium. (**D**) Ductal cell marker (KRT19) decreased in early passage (p-14) iSGEC-nSS2 to levels similar in late (p-80) cells. Conversely, ZO-1 and α-SMA expression decreased in later passaged (p-80) iSGECs with 1.2 mM Ca^2+^ and did not change in early (p-14) cells. Results calculated by Student’s *t*-test and *p*-values corrected by the Holm-Šídák multiple comparisons post hoc test (* *p* < 0.05). Error bars represent mean +/− SEM.

**Figure 6 jcm-09-03820-f006:**
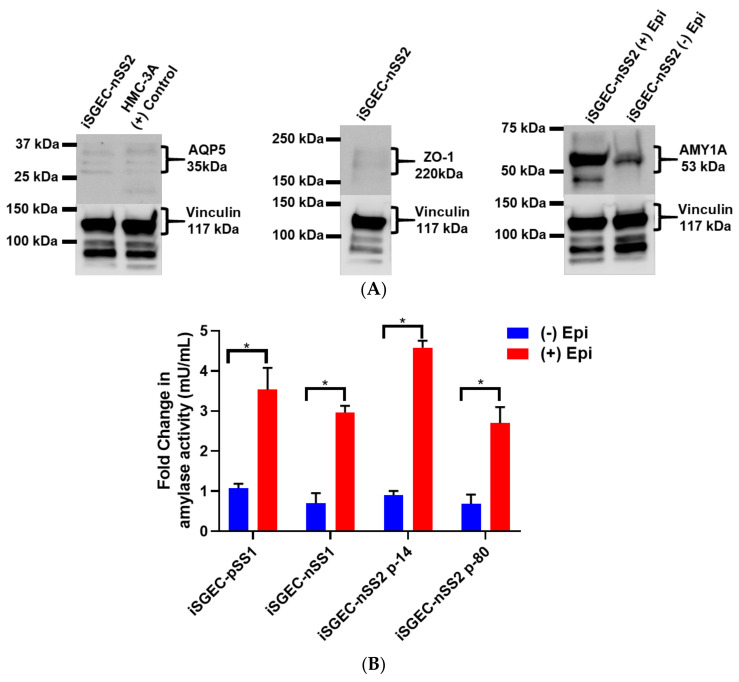
Protein expression of acinar and epithelial cell markers by Western blot in iSGEC-nSS2. **Legend.** (**A**) Protein expression of acinar (AQP5) (**left**) and epithelial (ZO-1) (**middle**) markers in whole-cell lysate of monolayer cultured iSGEC-nSS2. (**left**) Expression of AQP5 in iSGEC-nSS2 whole-cell lysate compared to HMC-3A, which was used as a (+) control for AQP5 in Western blots. (**right**) After stimulation with 10 µM epinephrine for 45 min ((+) Epi) cell culture supernatant was replaced, and cells were grown for 72 h before media were collected and concentrated. Cells without stimulation by media supplementation with 10µM epinephrine ((−) Epi) served as the control. Secreted α-amylase was detected in both stimulated ((+) Epi) and unstimulated ((−) Epi) cultures. 10µM epinephrine stimulated ((+) Epi) iSGEC-nSS2 cultures exhibited higher levels of α-amylase in cell culture supernatant than unstimulated cultures after 72 h. Western blots were normalized to vinculin expression in whole-cell lysate of their respective sample. (**B**,**C**) Media were harvested after stimulation with 10µM epinephrine for 45 min and amylase activity measured by colorimetric assay. (**B**) Supernatant of monolayer cultured iSGECs treated with 10µM ((+) Epi) had significantly higher α-amylase activity compared to untreated cells ((−) Epi). (C) Comparison of early (p-14) and late (p-80) passaged iSGEC-nSS2 secretion of α-amylase into media when cultured on matrigel. Results are listed as fold-change over the untreated control. Significance of differences between means was determined by Student’s *t*-test and *p*-values corrected using the Holm–Šidák multiple comparisons test (alpha = 0.05). Error bars represent mean +/− SEM. * *p* < 0.05.

**Figure 7 jcm-09-03820-f007:**
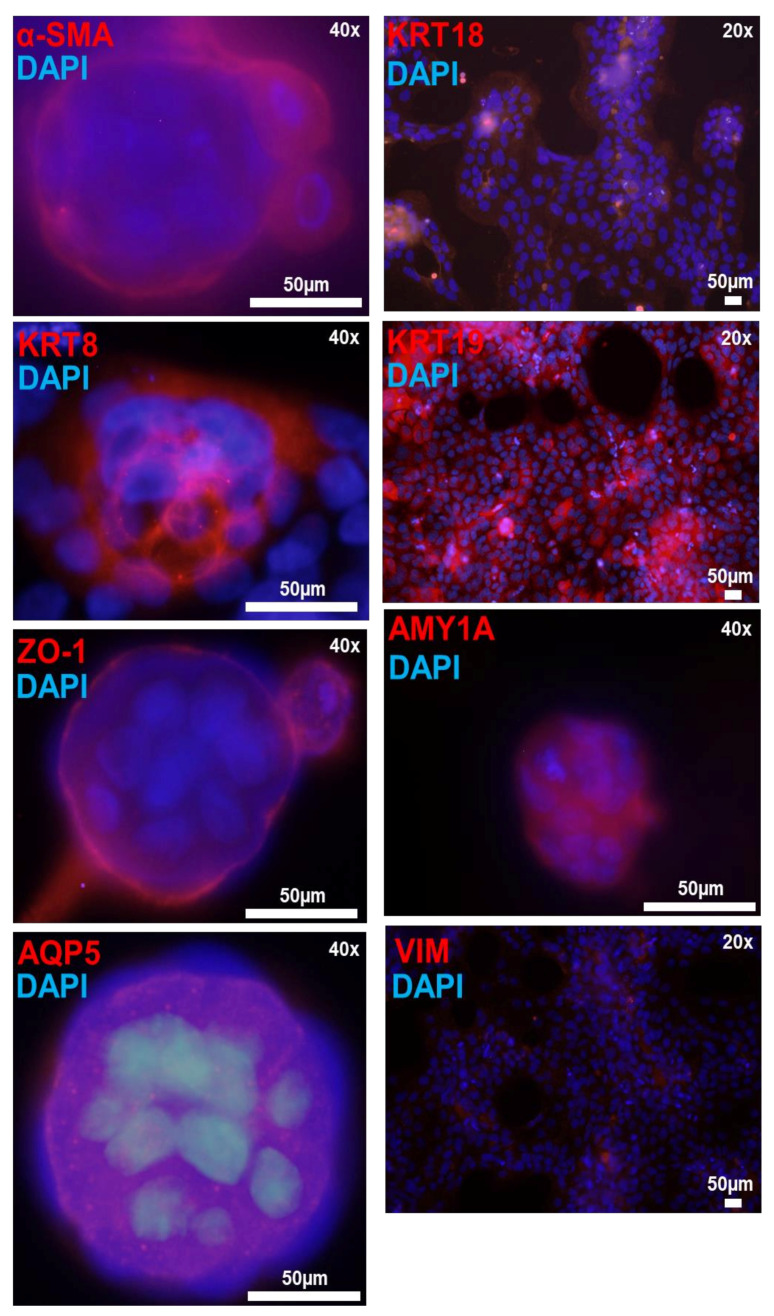
Immunofluorescence detection of salivary epithelium markers in matrigel cultured iSGEC-nSS2 cells. **Legend.** Detection of salivary epithelium markers (**red**) by immunofluorescence in matrigel cultures of iSGEC-nSS2 p-80 after 7 days. Spheroids expressed acinar cell markers (AQP5, AMY1A), tight-junction protein (ZO-1), and had clearly defined myoepithelial cells (α-SMA). Cells growing on the surface of the matrigel and spheroids expressed salivary epithelial markers KRT8, KRT18, and KRT19 proteins. Vimentin was expressed low and sparsely among cells and did not indicate regular staining patters. Scale bar represents 50 µm. DNA (**blue**) is highlighted with DAPI.

**Figure 8 jcm-09-03820-f008:**
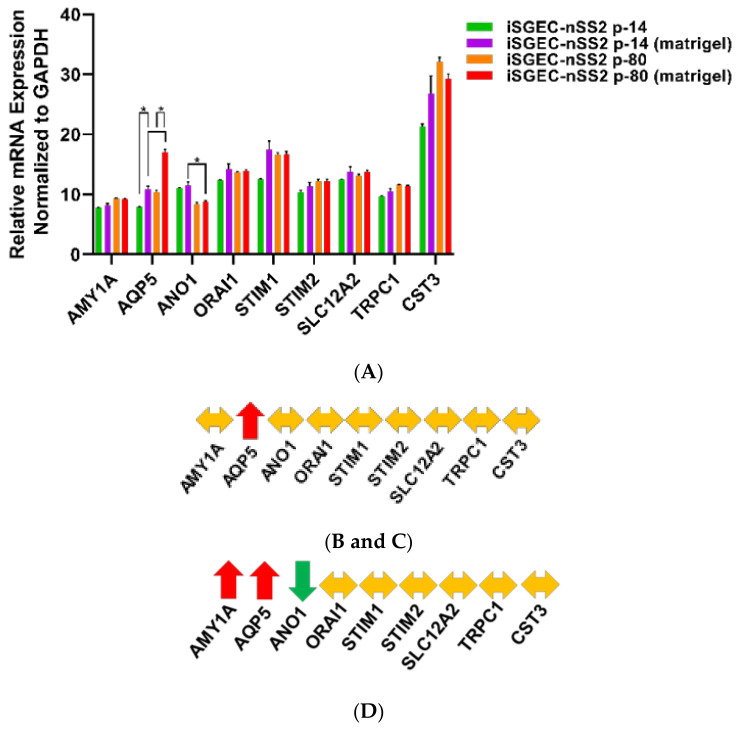
Changes in mRNA expression of early (p-14) and late (p-80) iSGEC-nSS2 cultures grown on matrigel. **Legend.** mRNA expression of early (p-14) and late (p-80) passaged iSGEC-nSS2 grown on either uncoated (monolayer) or matrigel coated plates by qRT-PCR (ΔCT). Changes in gene expression are indicated with arrows: increased (**red**), decreased (**green**), and no change (**yellow**). (**A**–**C**) AQP5 expression was the only acinar cell marker to increase in both p-14 (**B**) and p-80 (**C**) when cultured on matrigel (final: 2mg/mL) coated plates. In p-80 cells, matrigel had a greater effect on AQP5 expression compared to p-14. (**D**) Over extended passages, iSGEC-nSS2 expressed both AQP5 and AMY1A higher, whereas ANO1 decreased in expression over time. (**E**–**H**) The mRNAs of characterization markers CLDN1, CDH1, VIM, and KRT19 were differentially expressed in both p-14 and p-80 cultures, respectively, when grown as a monolayer or matrigel coated plates. (**F**–**G**) Only the ductal cell marker, KRT19, was consistently downregulated by matrigel. (**F**–**G**) Vimentin (VIM) can be an indicator of dedifferentiation in SGECs and was increased (**red**) in both p-14 and p-80 matrigel cultures. Results calculated by Student’s *t*-test and *p*-values corrected by the Holm-Šídák multiple comparisons test (* *p* < 0.05). Error bars represent mean +/− SEM.

**Table 1 jcm-09-03820-t001:** Patient demographics and clinical features.

Demographics	iSGEC-pSS1	iSGEC-nSS1	iSGEC-nSS2
Age	70	57	47
Gender	Female	Female	Female
Race	C	C	C
Clinical Features	pSS	nSS	nSS
Focus Score	1.8	0.3	0.16
Anti-Ro (SSA)	(−)	(−)	(−)
Unstim. Salivary Flow (<1.5 mL/15 min)	0.66	0.66	0.06 *
Stimulated Salivary Flow (mL/min)	11.7	8.91	1.02 *
Schirmer’s (+/−)	NA	NA	(−)
DMARDs	(−)	(−)	(−)

Labial salivary gland biopsies used in this study were collected from one pSS and two sicca patients (nSS). All patients were negative (−) for serum Anti-Ro (SSA) and were not taking disease-modifying anti-rheumatic drugs (DMARDs). * The patient with the lowest focus score had the lowest salivary flow rates (unstimulated/stimulated). NA: Schirmer’s test was not performed due to patient objection or information not listed in electronic medical records.

**Table 2 jcm-09-03820-t002:** Immortalized salivary gland epithelial cells (iSGECs), salivary gland cell lines (SGCLs), and HeLa protein expression of characterization markers by immunocytochemistry (ICC).

Protein Target	KRT8	K18	K19	ZO-1		E-Cadherin		AQP5		Vimentin		α-Amylase		α-SMA
Cell Line														
iSGEC-pSS1	+	+	+	+	+	+	+	+	+	+	+	+	+	(−/+)
iSGEC-nSS1	+	+	+	(−/+)	(−/+)	+	+	+	+	+	+	+	+	(−/+)
iSGEC-nSS2	+	+	+	+	+	+	+	+	+	+	+	+	+	(−)
A253	+	+	+	(−/+)		+		(−/+)		+		+		+
HMC-3A	+	+	+	(−/+)		+		+		+		+		+
HSY	+	+	+	(−/+)		+		(−/+)		+		+		+
HSG	+	+	+	(−/+)		+		(−/+)		+		+		+
HeLa	+	+	+	(−/+)		(−)		(−/+)		+		(−/+)		+

Results of protein expression of characterization markers in iSGECs, SGCLs, and HeLa cells by ICC are shown. Cells were grown on type-1 collagen-coated coverslips for up to 72 h prior fixation. iSGECs were additionally plated on coverslips and grow in SGEC sub-culturing media supplemented with 1.2mM Ca^2+^ to determine changes in acinar cell marker expression. (+) indicates positive expression of protein. (+/−) indicates low or diffuse expression of protein. (−) indicates no protein expression was detected by methods outlined.

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
