# Peer review of "Immortalization of Salivary Gland Epithelial Cells of Xerostomic Patients: Establishment and Characterization of Novel Cell Lines"

_jcm, 2020, doi:10.3390/jcm9123820_

Round 1
Reviewer 1 Report
The manuscript is well written, but the study has some limitations. The immortalized SGEcs were analyzed only from three female patients with xerostomia symptoms using labial salivary gland biopsies, and only one of them was with primary Sjögren’s syndrome. I missed the data about patients’ age and clinical data. There can be many co-factors affecting the results of mRNA expression of specific epithelial and acinar cell markers, as e.g., age, gender, environment, drugs, etc. These possible confounders should be addressed and discussed.
Author Response
Reviewer 1
The manuscript is well written, but the study has some limitations. The immortalized SGECs were analyzed only from three female patients with xerostomia symptoms using labial salivary gland biopsies, and only one of them was with primary Sjögren’s syndrome.
- I missed the data about patients’ age and clinical data.
- There can be many co-factors affecting the results of mRNA expression of specific epithelial and acinar cell markers, as e.g., age, gender, environment, drugs, etc. These possible confounders should be addressed and discussed.
Our response: We appreciate this Reviewer’s comment. Information regarding the patients’ age and clinical data is provided in Table 1 of the manuscript. To address the Reviewer’s concern regarding confounding factors, we have added information about disease-modifying anti-rheumatic drugs (DMARDs) to the clinical information listed in Table 1. None of the patients who provided SG biopsies were taking DMARDs. In addition, we sought to develop salivary gland cell line culture models which better represent the population of patients with pSS since the disease mainly affects females with the onset of symptoms typically occurring at post-menopausal age. Having transformed successfully only three primary SG cell cultures, it would not be possible to determine the effects of these factors and obtain statistically meaningful data. However, to address this point, we have revised the “Limitations” sub-section to recognize that age and gender may intrinsically affect expression profiles of the iSGECs developed (see highlight).
Reviewer 2 Report
The authors have conducted an interesting in vitro experimental study to generate and characterize immortalized salivary gland epithelial cells derived from labial salivary gland biopsies of Sjögren’s Syndrome patients, using appropriate control cells. The experimental approach and methodology used by Nollet al. in this work are well designed and have produced high-quality data. The paper is well organized and well written. As clearly reported in the discussion, the experimental model developed in this study will also allow other scientists to study and clarifythe interactions between ductal and acinar cells within the salivary gland epithelium that seems to be a key point to identify the molecular mechanisms underlying chronic autoimmune pathologies such as Sjögren’s syndrome. The really important datum that comes from this work is to have set up a salivary gland cell culture from salivary gland epithelial cells of Sjögren’s syndrome patients that retained the ability to form spheroids and could be used as an alternative to currently available cell culture models.Obviously, as reported in the "Limitations" section, further experiments are necessary for a better molecular characterization of the cells in culture and this should encourage the authors to continue in this study.
There are some minor corrections to the figures to be done before the publication:
- In figure 1, the pictures have different colours and contrast and the scale bars are missing.
- In figure 7, the scale bar value is missing.
Author Response
There are some minor corrections to the figures to be done before the publication:
Our response: We are grateful to the comments of this reviewer, and also appreciate words of encouragement to pursue this work.
- In figure 1, the pictures have different colors and contrast and the scale bars are missing.
Our response: Images were taken at different time points and the white balance/ color was auto-configured by the Olympus imaging software. To reduce the differences in contrast and hue without compromising image quality/ depiction, we have now converted the images to greyscale.
- In figure 7, the scale bar value is missing.
Our response: We have added the scale bar values to each image for better clarity.
Reviewer 3 Report
Major points :
Authors established immortalized cell lines derived from female Sjogren's syndrom (SS) or non-SS xerostomic patients. Although the cell lines have features of epithelial and pro-acini cells, overall data are incompletely support authors' hypothesis, requiring further data to clearly demonstrate that established cell lines represent a model for salivary gland research.
- Related to Figure 3, 4, 5, 8
mRNA expressions of several markers were shown and compared between cell lines and passages. But I wonder these data are correctly compared with good controls. To compare gene expression between primary SS-derived cell lines and non-SS-derived cell lines, cell lines from more patients were assessed for gene expression. Also, comparison of gene expression with originated tissue would be a good control.
- Related to Figure 5.
What make differences between cell lines (p14) at early and late (p80) passages? Authors should discuss, in more details, the outcome of differential expressed genes and what is in a good condition when compared with early and late passaged cell lines.
- Related to Figure 2.
Did you check spheroid formation efficiency between cell lines and passages? It would be more interesting to assess differences in gene expression between 2D monolayer culture and 3D spheroid culture, because some 3D cultures have been shown to increase stemness or cellular heterogeneity. Authors should add the data or describe about these points.
- Figure 1
In figure. 1 C-D, you said that “cuboidal morphology was maintained”. However, early iSGEC-nSS2 looks more sharp and thin.
Minor points :
- All figures are too sparse, miss align with different font size and style, especially marker expression part.
- In figure 6A, ladder is missing. And figure legend is too short. At least, you should state that what is ‘(-)EPI/(+)EPI’ and difference between B and C. In figure 6A, addition of arrow would be helpful to indicate which bands are in proper size.
- Figure 6D and 6E are described is result sections but omitted in figure or figure legend.
- In figure 7, most of data were not well-focused, making hard to be interpreted. More clear and focused data are required. I don’t know that why scale of Immunofluorescence image in figure 7 is different.
- In figure 8, ‘B and C’ is only one figure.3.
Author Response
Major points:
Authors established immortalized cell lines derived from female Sjogren's syndrome (SS) or non-SS xerostomic patients. Although the cell lines have features of epithelial and pro-acini cells, overall data are incompletely support authors' hypothesis, requiring further data to clearly demonstrate that established cell lines represent a model for salivary gland research.
Our response: We would like to thank this Reviewer for his thorough review. The points are well taken and addressing them will significantly improve the manuscript.
- Related to Figure 3, 4, 5, 8: mRNA expressions of several markers were shown and compared between cell lines and passages. But I wonder these data are correctly compared with good controls. To compare gene expression between primary SS-derived cell lines and non-SS-derived cell lines, cell lines from more patients were assessed for gene expression. Also, comparison of gene expression with originated tissue would be a good control.
Our response: In Figure 3, we compared both acinar and characterization marker expression of our three cell lines to a variety of well-established salivary gland cell models. The purpose of this figure was to demonstrate that the expression profiles of our three cell lines were mirrored by common salivary gland cell models. These cell models served as a “basal level” comparative for distinguishing our cell lines as salivary-derived.
Culturing of SGECs has been common practice in salivary gland research for many years. However, there is a degree of controversy over the origin and whether these primary cultures are homogenous among different patients. Overall, by developing a series of iSGECs, we have provided a comparative reference for primary cultures of SGECs, thereby enabling greater consistency in future research.
In this publication, we did not intend to fully demonstrate whether our cell lines retained pSS or non-pSS expression characteristics, but to develop a more pertinent salivary gland cell culture model which retains the ability to form spheroids with relevant differentiated cell types.
In addition, comparison of immortalized cell lines with their origin tissue was not performed due the complex heterogeneity of salivary glands and the lymphocytic infiltration present in both non-SS and pSS patients. Moreover, we expect cells within the salivary gland environment to have altered expression profiles based on the extracellular environment, increased cytokine exposure (pSS), and potential lymphocyte presence. In the future (as mentioned in the discussion), it would be of benefit to utilize a technique such as laser capture assisted microdissection to determine in-situ expression profiles of the different cell types for comparative analysis.
- Related to Figure 5: what make differences between cell lines (p14) at early and late (p80) passages? Authors should discuss, in more details, the outcome of differential expressed genes and what is in a good condition when compared with early and late passaged cell lines.
Our response: We have added more details within the discussion section explaining the changes in differential expression of genes across early and late passages shown in both Figures 4 and 8 (see highlight).
- Related to Figure 2: did you check spheroid formation efficiency between cell lines and passages?
Our response: Spheroid formation efficiency was assessed subjectively. The full spheroid characterization (including formation efficiency at different passages) was beyond the scope of this study. More extensive characterization of the structural and functional properties of cultured spheroids is underway in our laboratory.
It would be more interesting to assess differences in gene expression between 2D monolayer culture and 3D spheroid culture, because some 3D cultures have been shown to increase stemness or cellular heterogeneity. Authors should add the data or describe about these points.
Our response: Comparison of gene expression among early and late iSGEC-nSS2 2D and 3D cultures was outlined in Figure 8. To address this concern, we have now included a more detailed interpretation within the Discussion and Limitations sections (see highlights).
- Figure 1: In figure 1 C-D, you said that “cuboidal morphology was maintained”. However, early iSGEC-nSS2 looks more sharp and thin.
Our response: A more appropriate word would have been “cobblestone” in this instance and has now been changed accordingly in Figure 1.
Minor points:
- All figures are too sparse, miss align with different font size and style, especially marker expression part.
Our response: Thanks for pointing this out. The figures were initially submitted as word files. We have now made font/image sizes adjustments.
- In figure 6A, ladder is missing. In figure 6A, addition of arrow would be helpful to indicate which bands are in proper size.
Our response: We have now added the molecular size ticks of ladder markers in Figure 6.
And figure legend is too short. At least, you should state that what is ‘(-)EPI/(+)EPI’ and difference between B and C.
Our response: The legend of Figure 6A has been revised accordingly to include a more detailed description. Additionally, the methods section has been changed accordingly to better reflect this figure (see highlight).
- Figure 6D and 6E are described is result sections but omitted in figure or figure legend.
Our response: Figures 6D and 6E were incorrectly referenced in the manuscript. This has now been corrected.
- In figure 7, most of data were not well-focused, making hard to be interpreted. More clear and focused data are required. I don’t know that why scale of Immunofluorescence image in figure 7 is different.
Our response: Thanks for bringing this up. In our revised version, we have now included higher quality images for Figure 7. Also, the scales of images vary due to the magnification levels among the different images. We have now included magnification markers on each image to more clearly indicate why the scale appears to change. To highlight the expression patterns of some cell markers more clearly, we chose to show a greater number of cells. Also, the flattened cell growth of some cells when cultured on matrigel formed a clear monolayer expressing epithelial protein markers, KRT18 and KRT19.
- In figure 8, ‘B and C’ is only one figure.3.
Our response: Figures 8B and C are presented as one image to reduce redundancy since both early and late passaged iSEGC-nSS2 cells behaved identically. We have included a more thorough explanation within the figure legend.
Reviewer 4 Report
Very interesting paper.
Authors focused on offering a novel stem cell line to study xerostomy patients. Several authors would say it is clinical standard to perform daily biopsies under local anesthesia without the need of such a model for that. Cell models for bioengineering of salivary gland offers multiple therapeutic in vitro possibilities that authors may address as a plus of this work.
Author Response
Our response: Thank you for the comments. We have revised the discussion accordingly.